# A national study of burnout and spiritual health in UK general practitioners during the COVID-19 pandemic

**Ishbel Orla Whitehead** *, **Suzanne Moffatt, Carol Jagger, Barbara Hanratty**

Population Health Sciences Institute, Faculty of Medical Sciences, Campus for Ageing and Vitality, Newcastle upon Tyne, United Kingdom

* orla.whitehead@newcastle.ac.uk

## Abstract

### Objectives

To quantify the burnout and spiritual health of general practitioners (GPs) in the United Kingdom (UK) who worked during the Covid-19 Pandemic.

### Design

Online survey, April/May 2021, distributed via emails to general practices, Clinical Commissioning Groups (CCGs), Health boards, Clinical Research Networks, professional groups, social media GP groups and networks.

### Setting

United Kingdom.

### Participants

1318 GPs who had worked in the National Health Service (NHS) during the COVID-19 pandemic (March 2020 –May 2021).

### Main outcome measures

Burnout scores, measured by the Maslach Burnout Inventory (MBI) for Medical Personnel; spiritual health, measured using the Functional Assessment of Chronic Illness Therapy—Spiritual Well-Being, Non-Illness (FACIT-SP-NI).

### Results

19% of surveyed GPs were at the highest risk for burnout, using accepted MBI 'cut off' levels. There was no evidence of a difference in burnout by gender, ethnicity, or length of service. GP burnout was associated with GP spiritual health, regardless of identification with a religion. GPs with low spiritual health were five times more likely to be in the highest risk group for burnout.

**Data Availability Statement:** Data cannot be shared publicly because participants were not specifically consented for this. Data are available from Orla Whitehead via Newcastle University for

researchers who meet the criteria for access to confidential data. This is human research participant data. Participants were asked to consent to anonymised data "becoming part of a data set which can be accessed by other users running other research studies at Newcastle University and in other organisations. These organisations may be universities, or NHS organisations. [This] information will only be used by organisations and researchers to conduct research." The authors are concerned that this doesn't include consent for public data sharing, only for further research in universities or NHS organisations. Data will be shared upon reasonable request to the authors. The sentence "Data Access: While participants were not consented to allow public sharing of this data, data is available upon reasonable request to the authors." has been added. The authors have noted the difficulties here, and will amend the consent process for the future, to allow data sharing more easily. Data available from: Whitehead, Ishbel; Hanratty, Barbara; Moffatt, Suzanne; Jagger, Carol. (2022): A National Study of Burnout and Spiritual Health in UK General Practitioners During the COVID-19 Pandemic. Newcastle University. Dataset. https://data.ncl.ac.uk/articles/dataset/A_National_Study_of_Burnout_and_Spiritual_Health_in_UK_General_Practitioners_During_the_COVID-19_Pandemic/20418519 rdm@ncl.ac.uk can be contacted if the authors are unavailable to gain access to the data for researchers who meet the criteria for access to confidential data (i.e. those engaged in ethically approved research).

**Funding:** OW is funded by the National Institute for Health Research (NIHR) on an in practice fellowship (NIHR301074). https://fundingawards.nihr.ac.uk/award/NIHR301074 BH is funded by the NIHR Applied Research Collaboration North East and North Cumbria. https://arc-nenc.nihr.ac.uk/ The views expressed are those of the authors and not necessarily those of the NIHR or the Department of Health and Social Care. The funders had no role in study design, data collection and analysis, decision to publish, or preparation of the manuscript.

**Competing interests:** The authors have declared that no competing interests exist.

## Conclusions

Burnout is at crisis levels amongst GPs in the UK NHS. A comprehensive response is required, identifying protective and precipitating factors for burnout. The potentially protective impact of spiritual health merits further investigation.

## Introduction

The Covid-19 pandemic has highlighted concerns about burnout in doctors across Europe [1, 2]. Burnout is understood to be an occupational phenomenon rather than mental or physical illness and has been described as an 'erosion of the soul' [3]. The World Health Organisation (WHO) has defined burnout as a syndrome resulting from chronic workplace stress that has not been successfully managed, characterized by feelings of exhaustion, increased mental distance from one's job and reduced professional efficacy [4]. Burnout was already thought to be contributing to current workforce crises, along with higher rates of hazardous drinking and suicide among doctors [5]. General Practitioners (GPs) are particularly vulnerable to burnout [6], and, as manpower levels fall and workload increases [7], burnout amongst remaining GPs becomes more likely [8].

Spiritual health has been linked with reduced risk of burnout in doctors and other groups [9, 10]. Definitions of 'spiritual health' provided by GPs in a recent online survey mirror the WHO framing of burnout [11]. Sixty nine percent of the 177 respondents described themselves as a spiritual person with spiritual health defined as self-actualisation and meaning, transcendence and relationships beyond the self, and expressions of spirituality- including religious practice, meditation or yoga. Self-actualisation included concepts of soul and adhering to personal ethical and moral codes. Meaning referred to personal meaning to life and relationships; transcendence to a concept of something beyond the observable/physical, and relationships with communities, friends, family, nature and/or the divine. The link between spiritual health and burnout in doctors has been explored in previous studies. Previous studies on the relationship between spiritual health and burnout have been vulnerable to response or sampling bias, used unvalidated instruments, analysed single domains of burnout or single domains of spiritual health in isolation, and conflated religion, and wider spiritual health [12–14]. Two studies from outside the UK included primary care doctors showing some association between spiritual health and personal accomplishment, and higher perceived stress associated with lower religious activity [9, 15]. While burnout in GPs has been quantified using the MBI-HSS previously [16], this study adds an up to date quantification of burnout levels during the heart of the pandemic, using robust scores. GPs appear to be at risk of an epidemic of 'erosion of the soul' [3], poor holistic health, and burnout [8]. Identifying whether spiritual health and burnout are related in UK GPs will potentially allow a novel view of research into organisational and individual interventions to improve GPs spiritual health, possibly mitigating the current workforce crisis.

This survey compares burnout and spiritual health scores in the GP population in the UK who have worked during the Covid-19 pandemic, aiming to generate robust data to better understand relationships between practitioner health, wellbeing and burnout.

### Public and patient involvement

Patients and the public were involved in the conception and design of this research, raising concerns over strains on primary care, and in the interpretation of the results.

## Participants and methods

### Measuring burnout

We used the Maslach Burnout Inventory Human Services Survey for Medical Personnel (MBI) [3], a measure of burnout [17]. This measure includes three burnout domains—depersonalisation (DP), emotional exhaustion (EE) and personal accomplishment (PA). High scores in DP, EE and low scores in PA are thought to indicate high risk of burnout. While the MBI authors and others caution against use of dichotomous 'cut-offs' within the scores [18], traditional 'cut offs' (DP>10, EE>27 and PA<33) have been used in other studies to denote high risk of burnout [19]. Participants were not blinded as to the topic of the study.

### Measuring spiritual health

We used the Functional Assessment of Chronic Illness Therapy—Spiritual Well-Being, non-illness version(FACIT-Sp-NI) measure of spiritual health, for use in a non-patient population [20]. The FACIT-Sp is a validated measure of spiritual wellbeing, judged as one of the two best in a systematic review [21]. Previous work led us to expect a significant group of GPs to be secular [11], with some GPs hostile to concepts of spiritual health, especially religious practice. The three domains in the scale (meaning, peace and faith) reflected GP definitions of 'spiritual health' from previous research [11], and was suitable for both religious and secular populations. As the MBI has 22 questions, it was required that the spiritual health measure be as concise as possible. We decided to assess spiritual wellbeing, rather than religious or spiritual coping, or spiritual distress. Spiritual wellbeing best addressed the question of the relationship between spiritual health and burnout. Religiosity or spiritual beliefs were not assessed. To allow comparison with the MBI, the spiritual health measure needed to be a similarly on the day measure. The FACIT-Sp-NI was used in October 2020 for a similar study comparing the MBI and spiritual health in the USA [22].

An online survey was written using JISC online surveys, using the MBI and the FACIT-SP-NI to measure burnout and spiritual health. It was advertised via email and social media to UK GPs who had worked since March 2020, for completion during April and May 2021. Ethical approval was given by Newcastle University on 2nd February 2021, and HRA approval 12th April 2021. Written consent was sought online as a condition for proceeding with the survey. Demographic information: age, gender, ethnicity, number of sessions (half a day, or 4–5 hours) worked as a GP, geographical area of work, number of years as a GP, country of primary medical qualification (PMQ), country of GP training; was requested to describe the sample and enable comparison with the wider GP population.

### Data analysis

The MBI cannot be analysed as a sum total 'burnout score', so each domain was split into high, moderate and low tertiles. Those in the highest tertile of depersonalisation (DP), emotional exhaustion (EE) and lowest tertile of personal accomplishment (PA) were considered to be at highest risk for burnout, and those in the lowest tertile of depersonalisation (DP), emotional exhaustion (EE) and highest tertile of personal accomplishment (PA) were considered to be lowest risk of burnout [23] (S2 and S3 Tables). Therefore, those in the intermediate risk category covered a broad spectrum of burnout risk.

Differences in mean DP, EE and PA scores by religion, country of PMQ, and ethnicity were assessed by the Kruskal-Wallis test, differences in burnout domain scores by gender, by Mann-Whitney test and differences in mean spiritual score by religion/no religion by Student's t test, and the relationship between religion and burnout by Chi squared tests. The relationship

**Table 1. Characteristics of respondents.**

| | Number (%) (n = 1318) | GP Workforce data from December 2018 [25] |
|---|---|---|
| **Gender** (as recorded by the General Medical Council) | | |
| Female | 869 (66%) | 53% |
| Male | 442 (34%) | 44% |
| **Ethnic Group** | | |
| White | 1072 (81%) | 53% |
| Asian or Asian British | 174 (13%) | 25% |
| Mixed / Multiple ethnic background | 34 (3%) | |
| Black, Black British, Caribbean or African | 15 (1%) | |
| Other background | 22 (2%) | |
| **Religion** ("What is your religion?" religious identity, practice is not assumed.) | | No data held for comparison for GPs.* |
| No religion | 514 (39%) | |
| Christian | 579 (44%) | |
| Muslim | 75 (6%) | |
| Hindu | 51 (4%) | |
| Humanist | 20 (2%) | |
| Buddhist | 14 (1%) | |
| Atheist | 13 (1%) | |
| Jewish | 11 (1%) | |
| Sikh | 8 (1%) | |
| Other | 30 (2%) | |
| **Area of current work** | | |
| Scotland | 169 (13%) | |
| Wales | 152 (12%) | |
| North East | 122 (9%) | |
| Yorkshire and the Humber | 110 (8%) | |
| South West | 103 (8%) | |
| North West | 89 (7%) | |
| South London | 70 (5%) | |
| East of England | 70 (5%) | |
| East Midlands | 64 (5%) | |
| West Midlands | 64 (5%) | |
| Northern Ireland | 57 (4%) | |
| North West London | 55 (4%) | |
| Thames Valley | 54 (4%) | |
| Wessex | 52 (4%) | |
| Kent, Surrey and Sussex | 48 (4%) | |
| North Central and East London | 38 (3%) | |
| Other | 1 (<1%) | |
| **Country of primary medical qualification** | | |
| England | 845 (64%) | |
| Scotland | 183 (14%) | UK- 78% |
| Wales | 103 (8%) | |
| Northern Ireland | 53 (4%) | |
| European Economic Area (EEA) | 48 (4%) | |
| International Medical Graduate (IMG) | 86 (7%) | EEA- 5% |
| **Country of GP training** | | IMG-17% |

*(Continued)*

**Table 1.** (Continued)

|  | Number (%) (n = 1318) | GP Workforce data from December 2018 [25] |
|---|---|---|
| England | 966 (73%) |  |
| Scotland | 168 (13%) |  |
| Wales | 129 (10%) |  |
| Northern Ireland | 46 (3%) |  |
| Elsewhere | 9 (<1%) |  |

* Data from the British Social Attitudes Survey in 2018 gives general UK population statistics: "Do you regard yourself as belonging to any particular religion?" 52% no religion, 38% Christian, and 9% non-Christian religions, however it is expected that the GP population would be more diverse, as GPs tend to be more ethnically diverse than the populations they serve [26].

between burnout and spiritual health scores adjusting for potential confounding factors such as gender, ethnicity, years working as a GP, number of sessions worked, country of PMQ and GP training, and religion was investigated by multinomial logistic regression.

Data analysis was conducted using the Stata SE 17.0 package [24].

## Results

In total 1320 general practitioners responded. Two were excluded due to an incomplete MBI (n = 1), and implausible responses (work pattern of 60 sessions/week) (n = 1), therefore the analytic sample comprised 1318 responses. There were few missing data: gender (n = 7), religion (n = 4), ethnic group (n = 3), with no overlap.

### Respondent characteristics

Compared to GP Workforce data, survey responses showed an underrepresentation of GPs who were male or from a minority ethnicity, and likely under-representation of those of non-Christian religions (Table 1). Most GPs in the sample worked between 5 and 8 sessions (Table 1), and this was used as the baseline group in further analyses.

### Burnout scores

Differences in burnout domains by gender, ethnicity, religion and country of PMQ were generally small although women had higher mean EE than men, Asian or Asian British participants had lower PA scores, and GPs who graduated in the European Economic Area (EEA) or other countries outside the UK had lower mean DP scores (Table 2). Number of sessions worked correlated weakly with all domains of the burnout score, however there was no difference in mean sessions worked for those at highest and lowest risk of burnout.

The median score for each burnout domain was compared with the cut offs for 'high risk' of burnout used in previous research [27] (Fig 1).

As participants disagreed on whether humanism should be classified as a religion, the 2% (n = 20) humanists were omitted to allow comparison of religious identification with no religion. Atheists were included in 'no religion.' Those identifying with a religion had a higher mean spiritual score, 30.20 (95%CI 24.25 to 25.86) compared with those identifying as not having a religion (25.01, 95%CI 29.47 to 30.94) (Table 3), this was the case for each domain (meaning, peace and faith) of the spiritual score. Identification with a religion did not affect classification as being at highest or lowest risk of burnout overall (Table 3).

**Table 2. Median burnout domain scores compared by gender, ethnicity, and country of primary medical education.**

| Gender | Depersonalisation (DP) | Emotional Exhaustion (EE) | Personal Accomplishment (PA) |
|---|---|---|---|
| | Median (IQR) | Median (IQR) | Median (IQR) |
| Males | 10 (5–17) | 33 (23–42) | 37 (32–42) |
| Female | 10 (5–16) | 36 (26–43) | 36 (31–41) |
| Mann-Whitney | U-0.21 | U = -2.58 | U = 2.35 |
| p value | p = 0.83 | p = 0.01 | p = 0.02 |
| **Ethnic group** | | | |
| White | 10 (5–16) | 35 (25–43) | 37 (32–41) |
| Asian or Asian British | 10 (6–16) | 35 (23–43) | 35 (30–40) |
| Black, Black British, Mixed or other ethnic group | 12 (6–20) | 35 (23–43) | 38 (31–42) |
| Kruskal–Wallis | $\chi^2 = 3.76$ | $\chi^2 = 0.19$ | $\chi^2 = 6.57$ |
| p value | p = 0.15 | p = 0.91 | p = 0.04 |
| **Religious/non-religious identity** (humanists excluded) | | | |
| Religion | 10 (5–16) | 34 (27–43) | 37 (31–41) |
| No religion | 11 (5–18) | 36 (27–43) | 36 (31–41) |
| Kruskal–Wallis | $\chi^2 = 4.95$ | $\chi^2 = 7.53$ | $\chi^2 = 0.15$ |
| | p = 0.02 | p<0.01 | p = 0.70 |
| **Country of primary medical education** | | | |
| UK (United Kingdom) graduate | 10 (5–17) | 35 (25–43) | 36 (32–41) |
| EEA (European economic area) graduate | 9.5 (3.5–15) | 33.5 (21–40.5) | 38.5 (31–43) |
| IMG (international medical graduate i.e. outwith the UK or EEA) | 8 (4–14) | 34.5 (23–41) | 36.5 (32–41) |
| Kruskal–Wallis | $\chi^2 = 5.40$ | $\chi^2 = 2.40$ | $\chi^2 = 1.10$ |
| (p value) | p = 0.07 | p = 0.30 | p = 0.58 |

IQR = interquartile range

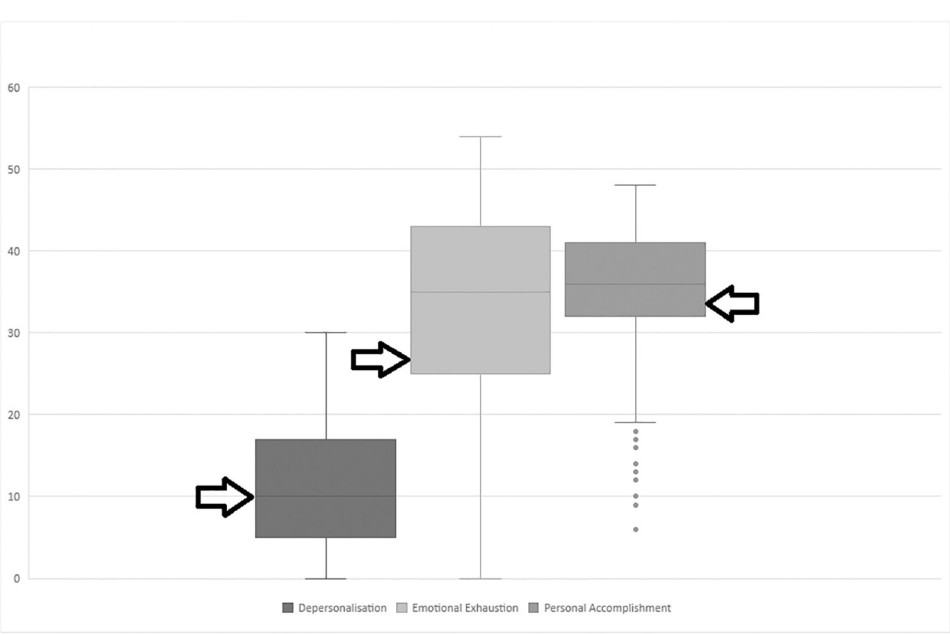

**Fig 1. Box plots of the scores by domain of the Maslach Burnout Inventory, with the commonly used 'cut off' for high risk for burnout scores marked.**

**Table 3. Burnout and spiritual scores compared for those who identified as having a religion, compared with those stating 'no religion' or atheist.**

| | Number of Participants | | |
|---|---|---|---|
| Burnout risk classification | No religion | Religion | Total |
| Lowest risk of burnout | 54 | 111 | 165 |
| Intermediate | 404 | 576 | 980 |
| Highest risk of burnout | 67 | 84 | 151 |
| Total | 525 | 771 | 1,296 |
| (Pearson chi2(2) = 5.2887 Pr = 0.07) | | | |
| Mean Total spiritual score (confidence interval) | 25.01 (24.25–25.86)* | 30.20 (29.47–30.94)* | t test for difference between means p<0.05 |

## Relationship between burnout and spiritual health

According to traditional 'cut offs' used in other studies [27], 256 participants (19%) had high DP, high EE and low PA defining a high risk of burnout. Cut offs based on tertiles suggested 48% had high depersonalisation, 68% had high emotional exhaustion and 30% had low personal accomplishment, with 12% (n = 155) falling into the high tertile for DP and EE, together with the low tertile for PA, and at the highest risk for burnout. The 13% (n = 169) who fell into the low tertile for DP and EE, as well as high for PA were considered lowest risk. All other participants were analysed as intermediate risk. The total spiritual score was similarly split into tertiles of high, moderate and low spiritual health.

In unadjusted analyses, participants with low spiritual health (compared to moderate) were five times more likely to be in the highest risk group for burnout than the intermediate group (RR = 5.09, 95%CI 3.33–7.78) (Table 4). After adjustment for gender, ethnicity, religion, years working as a GP, area of primary medical qualification and number of sessions worked, these associations persisted (RR = 5.46, 95%CI 3.52–8.46).

## Statement of principal findings

This large survey of UK GPs examined the relationship between burnout and spiritual health. We found that one in five GP participants were at very high risk of burnout using traditional cut off scores. Nearly half of all GPs scored highly on depersonalisation, two thirds scored highly on emotional exhaustion, and a third had low personal accomplishment. GPs with

**Table 4. Association between spiritual health score and risk of burnout: Multinomial logistic regression analysis.**

| Burnout Classification | Spiritual Health score | Unadjusted model | | Adjusted[1] | |
|---|---|---|---|---|---|
| | | Relative risk ratio [95% confidence interval] | p-value | Relative risk ratio [95% confidence interval] | p-value |
| **Lowest Risk** | | | | | |
| | Low | 0.25 [0.11–0.57] | 0.001 | 0.24 [0.11–0.56] | 0.001 |
| | Moderate | 1 (reference) | | 1 (reference) | |
| | High | 3.89 [2.63–5.75] | <0.001 | 3.99 [2.65–6.01] | <0.001 |
| **Intermediate risk** | | | | (reference) | |
| **Highest risk** | | | | | |
| | Low | 5.09 [3.33–7.78] | <0.001 | 5.46 [3.52–8.46] | <0.001 |
| | Moderate | 1 (reference) | | 1 (reference) | |
| | High | 0.31[0.14–0.68] | 0.003 | 0.24 [0.10–0.56] | 0.001 |

[1]Adjusted for gender, ethnic group, religion, years of work, number of sessions, and country of primary medical qualification.

lower levels of spiritual wellbeing were more likely to be at higher risk of burnout whilst higher levels of spiritual wellbeing appear to be protective against burnout. The relative risk of being most liable to burnout is five times higher for GPs who score lower on the FACIT-SP-NI score of spiritual wellbeing. The risk of being in the lowest risk group for burnout is nearly four times higher for those with a higher score on the FACIT-SP-NI.

Burnout appears to be a potential problem for all GPs. Overall, the effects of gender, length of service, ethnicity, and religion were small. Doctors who graduated outside the UK appear to have lower depersonalisation scores, and a lower risk for burnout. The number of sessions worked was positively correlated with both DP and EE scores, but the association with burnout risk did not reach statistical significance. High burnout levels are concerning for the wellbeing of GPs and their patient populations, with implications for the workforce, economic costs, and patient safety [6].

## Strengths and weaknesses

This study is the first within the UK to use validated measures of both burnout and spiritual wellbeing score in doctors. While the spiritual score included a faith domain, religiosity was not assessed. While the participants worked during the pandemic, no specific questions related to the pandemic- this study offers a mid-covid pandemic snapshot. Doctors from minority ethnic groups and international medical graduates were under-represented in this study. It is also likely that many participants had a particular interest in the topics, due to the volunteer nature of the sample. However, the consequences of possible sampling bias were mitigated by the number of participants, which was larger than previous studies [28], and from an often difficult to reach group.

## Comparison with other studies

This study has described greater EE, yet higher PA compared to previous studies. Twenty years ago, a study of 564 UK GPs (considered to be the largest sample of the MBI at the time) [28] described high EE in 46%; high DP in 42%; and low PA in 34%. Comparable surveys have described lower levels of burnout amongst GPs in other countries. In a 2018 sample of Danish GPs, 31% had high EE, 21% high DP and 37% low PA [29], similarly, German GPs surveyed in 2014, reported high EE in 34.1%, high DP in 29.0% and low PA in 21.5%, with 7.5% of their sample considered high risk for burnout [19]. French GPs in 2019 reported similar levels of EE and DP (23.8%, 27.3%) but fewer with low PA (13.3%) [30]. A recent systematic review of data prior to the pandemic found heterogeneity in approaches to analysis of the MBI-HSS, as well as a wide variation in burnout prevalence, and lower pooled burnout scores on meta-analysis [16]. None of these studies measured spiritual health. In the UK, the most recent (10th) GP worklife survey from 2019 showed that job satisfaction for GPs had improved slightly since 2017 (but not to previous levels) [31]. Other surveys of doctors found a perception that burnout had risen during the pandemic [32], despite already being at intolerable levels.6 A systematic review of GP burnout during the pandemic found two European studies that found 24.5% to 46.1% GPs had emotional exhaustion [33]. A recent survey by the British Medical Association reported that over half of UK doctors who responded felt the term moral injury resonated with them [34], particularly for those from an ethnic minority [34]. This study does not show that gender affects risk of being at high risk of burnout, in contrast with other studies identified in a recent systematic review, which found female doctors more at risk of stress [33]. There is a dearth of data from UK GPs using the 'gold standard' measure of burnout, as shortened versions of the MBI or self-rating have been more commonly used [35].

The authors have found no published studies to date that measure burnout and spiritual health simultaneously in doctors using the validated measures used here (MBI and the FACIT-Sp-NI). An unpublished survey of 44 doctors (residents and residency faculty) in Kansas, USA using these measures reported a significant negative correlation between the DP and EE domains of the MBI and the FACIT score, and positive correlation between PA and the FACIT score [22]. The qualitative themes in their work included connection to others, camaraderie, empathy, and the use of religious coping [22]. Other research included a study of doctors at a 2013 US conference, which found MBI domains were correlated with the Hatch Spiritual Involvement and Beliefs score (SIBS) [36]. The same authors have reported positive (r = 0.35) correlations between the SIBS score and the PA domain of the MBI amongst residency doctors in the USA [15].

## Implications for policy and practice

There is considerable attention on burnout amongst GPs [16, 37], primarily focussing on workplace pressure, appointments, retirement and recruitment issues, increasing workloads over time, and the multiple challenges of the COVID-19 pandemic [16, 37]. Concern and suggestions of remedies for burnout are not new. Almost two decades ago, Chambers made three suggestions to reduce burnout, including managing trainee expectations, reducing the stigma of mental illness, and reducing isolation [38], all three of which remain pertinent today. The rise in burnout suggests that there is merit in looking at this issue from different perspectives. The traditional approaches of resilience, and mental and physical health support have failed to stem the rising tide of GP burnout. Increasing focus on the role of spiritual health, and the relationship between spiritual health and burnout, may offer new insights into how to improve the health and wellbeing of GPs. Destigmatisation of both burnout and spiritual distress may also be important. It is timely and important, to explore the relationship between burnout and spiritual health during the pandemic, to inform and protect the workforce during and after this crisis.

Future work could usefully consider the wellbeing of the wider team in primary care, including practice managers, as they may also be liable to burnout. GPs experiences of burnout and spiritual healthcare currently being explored using qualitative methods.

## Conclusion

This study gives evidence that high levels of concern about UK primary care are justified. GPs appear to be at higher risk of burnout than ever before. Decades of reporting on burnout has recorded increasing levels. Further research into GP spiritual health, and how this relates to burnout, may offer a breakthrough, to help improve the life and work of GPs, and keep them well in work through difficult times.

## Supporting information

**S1 Table. Comparison of burnout classification and spiritual score: Adjusted multinomial regression.**
(DOCX)

**S2 Table. Classifications of burnout risk for analysis used for this study.**
(DOCX)

**S3 Table. Burnout tertile 'cut offs' used in analysis of this data, with cut offs used in literature for comparison.**
(DOCX)

## Acknowledgments

Thanks to Voice North PPI group, whose discussion prompted this research, and to all the General Practitioners who took the time to take part, and help distribute the survey, along with other primary care staff. Thanks to Mindgarden and FACIT for use of their licenced tools. Thanks to Holly Bennett for her support with using the Stata package, and Daniel Stowe for statistical support.

## Author Contributions

**Conceptualization:** Ishbel Orla Whitehead, Suzanne Moffatt, Carol Jagger, Barbara Hanratty.

**Data curation:** Ishbel Orla Whitehead, Carol Jagger, Barbara Hanratty.

**Formal analysis:** Ishbel Orla Whitehead.

**Funding acquisition:** Ishbel Orla Whitehead, Suzanne Moffatt, Carol Jagger, Barbara Hanratty.

**Investigation:** Ishbel Orla Whitehead.

**Methodology:** Ishbel Orla Whitehead, Suzanne Moffatt, Carol Jagger, Barbara Hanratty.

**Project administration:** Ishbel Orla Whitehead, Suzanne Moffatt, Barbara Hanratty.

**Resources:** Ishbel Orla Whitehead.

**Software:** Ishbel Orla Whitehead.

**Supervision:** Suzanne Moffatt, Carol Jagger, Barbara Hanratty.

**Validation:** Ishbel Orla Whitehead, Carol Jagger, Barbara Hanratty.

**Visualization:** Ishbel Orla Whitehead.

**Writing – original draft:** Ishbel Orla Whitehead.

**Writing – review & editing:** Ishbel Orla Whitehead, Suzanne Moffatt, Carol Jagger, Barbara Hanratty.

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
