## [Decision Letter · Decision Letter 0]

8 Jun 2022

PONE-D-22-09960A National Study of Burnout and Spiritual Health in UK General Practitioners During the COVID-19 PandemicPLOS ONE

Dear Dr. Whitehead,

Thank you for submitting your manuscript to PLOS ONE. After careful consideration, we feel that it has merit but does not fully meet PLOS ONE’s publication criteria as it currently stands. Therefore, we invite you to submit a revised version of the manuscript that addresses the points raised during the review process.

Please make sure to address the reviewers' comments. 

We look forward to receiving your revised manuscript.

Kind regards,

Zhuo Chen, Ph.D.

Academic Editor

PLOS ONE

Journal Requirements:

Reviewers' comments:

Reviewer's Responses to Questions

**Comments to the Author**

1. Is the manuscript technically sound, and do the data support the conclusions?

Reviewer #1: Partly

Reviewer #2: No

2. Has the statistical analysis been performed appropriately and rigorously? 

Reviewer #1: Yes

Reviewer #2: No

3. Have the authors made all data underlying the findings in their manuscript fully available?

Reviewer #1: No

Reviewer #2: No

4. Is the manuscript presented in an intelligible fashion and written in standard English?

Reviewer #1: Yes

Reviewer #2: Yes

5. Review Comments to the Author

Reviewer #1: This paper addresses an important issue and has relevant findings. I present my concern for its methodology and presentation here:

Table 1: what is the difference between “no religion” and “atheist” here? Where will the agnostics (sometimes estimated to be 15% of the UK population) go? Given the very important role of this variable in this entire study, a better explanation of these categories is needed and more discussion is needed if some of frequencies deviate too much from previously estimated frequencies in comparable populations.

Line 197: so “atheists” are included as one of the religions???

Table format is fairly unconventional in this paper and Table 4 starts to get confusing. I suggest that the authors take a look at the published tables in PLOS series journals to better follow the expected style of presenting a multinomial logistic regression. Moreover, readers will expect to see what are the covariates used in the multinomial logistic regression, at least in the footnotes to Table 4 . Tables need to be self-explanatory, without having the readers go back to the text to check what are the covariates used.

As spirituality could mean very different things for religious people and non-religious people, I suggest that the multinomial model be run among the religious and then among the non-religious, separately. In other words, using religion as an effect modifier rather than a confounder will make more sense here. Alternatively, the authors could create an interaction term that multiplies the spirituality variable with the binary religion variable.

The religion variable measures the respondent’s identification with a specific faith group, which is helpful. Yet, this measurement does not get to the level of religiosity, i.e., how important is this faith to the individual, frequency of religious practice/service, etc. This aspect of religiosity could influence mental health outcomes more directly rather than the categories of faith groups or the binary classification of religion vs. no-religion, and thus should be discussed as a limit in Discussion under the context of previous findings.

Reviewer #2: Thank you very much for the opportunity to review the manuscript entitled “A National Study of Burnout and Spiritual Health in UK General Practitioners During the COVID-19 Pandemic”. GP burnout during COVID-19 is a profound challenge that needs to be addressed. However, I am afraid that I have the following concerns about the novelty of the study topic, soundness of research design, and depth of discussion:

1. Introduction: the authors stated that “Previous studies on the relationship between spiritual health and burnout have been vulnerable to response or sampling bias, used unvalidated instruments, analysed single domains of burnout or spiritual health in isolation, and conflated religion, and wider spiritual health” (p4 lines 88-91). As far as I understand, this statement is closely linked to the novelty of this study. However, I am concerned about two issues. First, the statement was made without any reference. Second, this study also suffered from some limitations such as response or sampling bias, analyzed spiritual health in isolation.

2. Another issue that is closely related to the novelty of this study is the theoretical/practical importance of understanding the relationship between spiritual wellness and burnout. However, I am afraid that I did not find strong justification in the introduction section.

3. The authors stated that “While burnout in GPs has been quantified using the MBI-HSS previously, this study adds an up to date quantification of burnout levels during the heart of the pandemic, using robust scores.” (p5 lines 93-95). I agree that quantification of burnout levels amongst GPs during pandemic is critical to help understand the psychological consequence of COVID-19. However, the prevalence/degree of burnout was neither part of the research objective “aiming to generate robust data to better understand relationships between practitioner health, wellbeing and burnout” (p5 lines 99-100), nor discussed in depth with prior studies that investigated burnout during COVID-19.

4. Participant recruitment: it seems that the authors adopted convenience sampling technique to include GPs who had worked between March 2020 and May 2021. What were the start and end dates of data collection? Besides the sample representativeness issues, a year seems to be very long for a cross-sectional study, considering that there were several waves of COVID during 2020 – 2021.

5. Statistical analysis: the responses of burnout do not seem to be in normal distribution (Figure 1). In this case, I do not suggest reporting mean values or conducting t test/ANOVA. I did not quite understand why not to control for areas of current work in regression analysis.

6. Discussion: I wish I could get more insights from the findings by comparing them with prior studies that investigated burnout during COVID-19.

---

## [Author Response · Author response to Decision Letter 0]

2 Aug 2022

We would like to thank the two reviewers who made thoughtful and useful suggestions for our paper titled “A National Study of Burnout and Spiritual Health in UK General Practitioners During the COVID-19 Pandemic”.

Please find our responses to each of the requested changes below.

Thank you for these clear instructions, the manuscript has been re-formatted. 

Apologies for this omission-“Written consent was sought online as a condition for proceeding with the survey” has been added to the methods and the ethics statement has been revised. 

This is human research participant data. Participants were asked to consent to anonymised data “becoming part of a data set which can be accessed by other users running other research studies at Newcastle University and in other organisations. These organisations may be universities, or NHS organisations. [This] information will only be used by organisations and researchers to conduct research.” The authors are concerned that this doesn’t include consent for public data sharing, only for further research in universities or NHS organisations. Data will be shared upon reasonable request to the authors. The sentence “Data Access: While participants were not consented to allow public sharing of this data, data is available upon reasonable request to the authors.“ has been added. The authors have noted the difficulties here, and will amend the consent process for the future, to allow data sharing more easily.

The ethics statement has been moved to the Methods section, and the previous ethics statement has been deleted.

This has been amended

Reviewers' comments:

Reviewer's Responses to Questions

Comments to the Author

1. Is the manuscript technically sound, and do the data support the conclusions?

Reviewer #1: Partly

Reviewer #2: No

2. Has the statistical analysis been performed appropriately and rigorously? 

Reviewer #1: Yes

Reviewer #2: No

3. Have the authors made all data underlying the findings in their manuscript fully available?

Reviewer #1: No

Reviewer #2: No

4. Is the manuscript presented in an intelligible fashion and written in standard English?

Reviewer #1: Yes

Reviewer #2: Yes

5. Review Comments to the Author

Reviewer #1: This paper addresses an important issue and has relevant findings. I present my concern for its methodology and presentation here:

Table 1: what is the difference between “no religion” and “atheist” here? Where will the agnostics (sometimes estimated to be 15% of the UK population) go? Given the very important role of this variable in this entire study, a better explanation of these categories is needed and more discussion is needed if some of frequencies deviate too much from previously estimated frequencies in comparable populations.

Participants were asked “what is your religion?” and to choose from “No religion, Christian, Muslim, Jewish, Sikh, Buddhist, Humanist, Hindu, or Other (please write).” This was left open to the participant to interpret, as to whether this is a cultural, ethnic, philosophical, or religiosity label. This study looks at spiritual health (which often involves religious aspects) as defined by the population being studied, and while religious identification was asked as a potential confounding variable, it was not the variable under investigation. 

Those who wrote in the “other” box were either left as “other” where they described being of mixed religious background, or from a smaller religious group. Those who named a subgroup of the main religions were analysed as part of that main religion. The authors debated how to analyse Humanists, as in a previous survey by the authors participants regarded Humanism as ‘a religion’, yet others stated Humanism is a lack of religion, hence the analysis of the religion variable being conducted with the small number of Humanists excluded. The nature of quantitative work in an area that is so dependent on personal philosophy is challenging, as in many ways we are looking to measure the immeasurable.

Line 197: so “atheists” are included as one of the religions??? 

The nature of atheism is also debatable- in our study, atheists were included in ‘no religion’, as were agnostics unless they named a religion they were agnostic about. There is an argument, put by a participant, that atheism is “stronger” than “no religion”, and is a belief system in itself. “Atheists were included in ‘no religion.’” has been added to the text to clarify where atheists were placed. “or atheist” has been added at line 207.

Table format is fairly unconventional in this paper and Table 4 starts to get confusing. I suggest that the authors take a look at the published tables in PLOS series journals to better follow the expected style of presenting a multinomial logistic regression. Moreover, readers will expect to see what are the covariates used in the multinomial logistic regression, at least in the footnotes to Table 4 . Tables need to be self-explanatory, without having the readers go back to the text to check what are the covariates used.

Thank you for your feedback on table 4. Lines 232-233 are the footnote which gives the covariates used. We believe the table is self-explanatory. The supplementary table (table S1) gives all unadjusted relative risks. As the main research question for this paper is regarding any association between the spiritual health score, and the burnout score, the regression analysis presented aims to answer that question.

As spirituality could mean very different things for religious people and non-religious people, I suggest that the multinomial model be run among the religious and then among the non-religious, separately. In other words, using religion as an effect modifier rather than a confounder will make more sense here. Alternatively, the authors could create an interaction term that multiplies the spirituality variable with the binary religion variable.

We agree, spirituality means different things for many groups of people. The authors spent time considering the spiritual health scores and believe that the FACIT-Sp-NI is equally applicable to those of any degree of religiosity. This study doesn’t measure religiosity, and the religion question was left open for interpretation for the participant, and therefore could well be part of ethnic or cultural identification rather than active religious participation. While a longer survey could have ‘drilled down’ into religiosity, religious practice, cultural membership of a faith, etc, the survey would likely not have the response rate it did if it had. The outcome of interest was the spiritual health score, the FACIT-Sp-NI, which does include a ‘faith’ subdomain in the score. The questions in the faith subdomain refers to ‘faith or spiritual beliefs’, rather than religiosity, and was included in the total FACIT-Sp-NI score analysed. Data were gathered around religious identification to identify any sampling bias towards only those with a religious identification for example. Authors previously have argued that spiritual health and religiosity overlap so much as to be the same (eg. Harold Koenig), however the definition we use, given by the population under study (UK GPs) in a previous survey, describes religious and spiritual behaviour as one aspect of overall spiritual health, similarly to Puchalski et al’s consensus definition. The data in table 3 does show that those who named a religion had significantly higher spiritual health scores than those who didn’t, which would be expected given the faith subdomain. Burnout scores were not statistically significantly affected by identification with a religion. (Table 3) As the FACIT-Sp-NI does have a faith subdomain, this could be analysed as a measure of religiosity, however the research question is regarding general spiritual health, and we have deliberately drawn no conclusions about religiosity as an effect modifier- this study is asking about the wider concept of spiritual health. Previous studies have addressed religiosity and burnout, finding mixed results. 

The religion variable measures the respondent’s identification with a specific faith group, which is helpful. Yet, this measurement does not get to the level of religiosity, i.e., how important is this faith to the individual, frequency of religious practice/service, etc. This aspect of religiosity could influence mental health outcomes more directly rather than the categories of faith groups or the binary classification of religion vs. no-religion, and thus should be discussed as a limit in Discussion under the context of previous findings.

Thank you, “While the spiritual score included a faith domain, religiosity was not assessed.” has been added in the strengths and limitations section for clarity. In our study, we did not measure religiosity.

Reviewer #2: Thank you very much for the opportunity to review the manuscript entitled “A National Study of Burnout and Spiritual Health in UK General Practitioners During the COVID-19 Pandemic”. GP burnout during COVID-19 is a profound challenge that needs to be addressed. However, I am afraid that I have the following concerns about the novelty of the study topic, soundness of research design, and depth of discussion:

1. Introduction: the authors stated that “Previous studies on the relationship between spiritual health and burnout have been vulnerable to response or sampling bias, used unvalidated instruments, analysed single domains of burnout or spiritual health in isolation, and conflated religion, and wider spiritual health” (p4 lines 88-91). As far as I understand, this statement is closely linked to the novelty of this study. However, I am concerned about two issues. First, the statement was made without any reference. Second, this study also suffered from some limitations such as response or sampling bias, analyzed spiritual health in isolation.

Thank you for highlighting these, references for the statement have been added. The sentence has been changed to “Previous studies on the relationship between spiritual health and burnout have been vulnerable to response or sampling bias, used unvalidated instruments, analysed single domains of burnout or single domains of spiritual health in isolation.” to clarify that other studies have looked at single domains (eg. Spiritual activities, use of spiritual coping, etc) rather than the use of a validated broad spiritual measure like the FACIT-Sp-NI. 

2. Another issue that is closely related to the novelty of this study is the theoretical/practical importance of understanding the relationship between spiritual wellness and burnout. However, I am afraid that I did not find strong justification in the introduction section.

Thank you for highlighting this. “Identifying whether spiritual health and burnout are related in UK GPs will potentially allow a novel view of research into organisational and individual interventions to improve GPs spiritual health, possibly mitigating the current workforce crisis.” This sentence has been added to the introduction.

3. The authors stated that “While burnout in GPs has been quantified using the MBI-HSS previously, this study adds an up to date quantification of burnout levels during the heart of the pandemic, using robust scores.” (p5 lines 93-95). I agree that quantification of burnout levels amongst GPs during pandemic is critical to help understand the psychological consequence of COVID-19. However, the prevalence/degree of burnout was neither part of the research objective “aiming to generate robust data to better understand relationships between practitioner health, wellbeing and burnout” (p5 lines 99-100), nor discussed in depth with prior studies that investigated burnout during COVID-19.

As far as the authors are aware, there have been no other national GP surveys in the UK using the ‘gold standard’ measure of burnout in populations, the Maslach Burnout Inventory – Human Services Survey. Jefferson et al1’s systematic review found one survey using the Perceived Stress Scale (not a burnout measure), and was limited to Leicester, rather than a national survey.2 Reference to Jefferson et al’s systematic review has been added to the comparison with other literature: “ A systematic review of GP burnout during the pandemic found two European studies that found 24.5% to 46.1% GPs had emotional exhaustion.”

4. Participant recruitment: it seems that the authors adopted convenience sampling technique to include GPs who had worked between March 2020 and May 2021. What were the start and end dates of data collection? Besides the sample representativeness issues, a year seems to be very long for a cross-sectional study, considering that there were several waves of COVID during 2020 – 2021.

The survey was conducted during April and May 2021. (line 48)

5. Statistical analysis: the responses of burnout do not seem to be in normal distribution (Figure 1). In this case, I do not suggest reporting mean values or conducting t test/ANOVA. I did not quite understand why not to control for areas of current work in regression analysis.

As the sample was large, it was felt that the type 1 error using t test and ANOVA would be insignificant. To remedy any concerns, the table has been re-written to contain median values, IQR and used the Mann-Whitney and Kruskal-Wallis tests.

There was no reason to hypothesise that burnout scores during this period would vary by broad locality currently worked in. There was no statistically significant difference in burnout scores by locality currently worked in. This data was gathered to ensure an adequate spread of recruitment, from all four nations within the UK. There were more participants from England, being the largest of the nations, and therefore comparing the smaller nations to England, could have led to misleading/erroneous conclusions by the reader about which area has GPs at lower/higher risk of burnout.

6. Discussion: I wish I could get more insights from the findings by comparing them with prior studies that investigated burnout during COVID-19.

This would be ideal, however the novelty of our study is that there aren’t similar published studies investigating burnout using validated scores in this population during COVID-19. We have added a sentence comparing our findings with those in Jefferson et al’s systematic review: “This study does not show that gender affects risk of being at high risk of burnout, in contrast with other studies identified in a recent systematic review, which found female doctors more at risk of stress.”

1. Jefferson L, Golder S, Heathcote C, et al. GP wellbeing during the COVID-19 pandemic: a systematic review. British Journal of General Practice 2022; 72: e325. DOI: 10.3399/BJGP.2021.0680.

2. Trivedi N, Trivedi V, Moorthy A, et al. Recovery, restoration, and risk: a cross-sectional survey of the impact of COVID-19 on GPs in the first UK city to lock down. BJGP Open 2021; 5: BJGPO.2020.0151. DOI: 10.3399/BJGPO.2020.0151.

---

## [Decision Letter · Decision Letter 1]

10 Oct 2022

PONE-D-22-09960R1A National Study of Burnout and Spiritual Health in UK General Practitioners During the COVID-19 PandemicPLOS ONE

Dear Dr. Whitehead,

Thank you for submitting your manuscript to PLOS ONE. After careful consideration, we feel that it has merit but does not fully meet PLOS ONE’s publication criteria as it currently stands. Therefore, we invite you to submit a revised version of the manuscript that addresses the points raised during the review process.

We look forward to receiving your revised manuscript.

Kind regards,

Zhuo Chen, Ph.D.

Academic Editor

PLOS ONE

Journal Requirements:

:

Reviewers' comments:

Reviewer's Responses to Questions

**Comments to the Author**

1. If the authors have adequately addressed your comments raised in a previous round of review and you feel that this manuscript is now acceptable for publication, you may indicate that here to bypass the “Comments to the Author” section, enter your conflict of interest statement in the “Confidential to Editor” section, and submit your "Accept" recommendation.

Reviewer #1: All comments have been addressed

2. Is the manuscript technically sound, and do the data support the conclusions?

Reviewer #1: Yes

3. Has the statistical analysis been performed appropriately and rigorously? 

Reviewer #1: Yes

4. Have the authors made all data underlying the findings in their manuscript fully available?

Reviewer #1: No

5. Is the manuscript presented in an intelligible fashion and written in standard English?

Reviewer #1: Yes

6. Review Comments to the Author

Reviewer #1: Overall the authors have done a decent job in addressing the comments. The one major concern I have for this version is the assertion in Line 277 "There are no published studies that measures burnout and spiritual health simultaneously using the validated measures (MBI and the FACIT-Sp-NI)". This is by far too strong a statement and I believe that studies like Akbari et al (2018) needs to be cited and discussed as part of the research context:

Akbari M, Hossaini SM. The Relationship of Spiritual Health with Quality of Life, Mental Health, and Burnout: The Mediating Role of Emotional Regulation. Iran J Psychiatry. 2018 Jan;13(1):22-31. PMID: 29892314; PMCID: PMC5994229.

I suggest that the authors consider and discuss more studies like Akbari et al (2018) in either Introduction or Discussion, and tone down the assertion from "There are no published studies" to "There have been relatively few studies".

7. PLOS authors have the option to publish the peer review history of their article (what does this mean?). If published, this will include your full peer review and any attached files.

Reviewer #1: No

---

## [Author Response · Author response to Decision Letter 1]

11 Oct 2022

Response to second review:

Thank you for very much for taking the time to review our manuscript. We value your feedback.

Thank you for highlighting this interesting study, I wonder if you’re one of the authors? This study adds to that evidence base, and does use validated tools to measure burnout and spiritual health. However, it does not use the FACIT-Sp-NI, which our study uses. We are aware that there are multiple studies linking the concepts of spiritual health and burnout in other professions, the example in this study is university staff. Thank you for highlighting some confusion in the message of lines 287 onwards- we were specifically referring to studies in doctors. The authors have recently conducted a (very recently updated) systematic review of the literature of studies measuring both spiritual health and burnout in doctors, (under review) and there were no studies using the FACIT-Sp-NI and MBI in doctors, apart from Schmitt et al’s abstract. We do not feel Akbari et al is the right study to be cited here, as the sample does not appear to include general practitioners, or even medical doctors, as their primary sample, and other more comparable studies have been chosen for comparison. The studies found in our systematic review that used validated burnout and spiritual health measurements were Doolittle et al (2013) and Doolittle et al (2015). Studies such as Antonsdottir I, Rushton CH, Nelson KE, Heinze KE, Swoboda SM, Hanson GC. Burnout and moral resilience in interdisciplinary healthcare professionals. Journal of Clinical Nursing. 2022;31(1):196-208. could be comparable, as a recent study, however this study only compares religion with burnout, rather than the wider spiritual health, Salmoirago-Blotcher E, Fitchett G, Leung K, Volturo G, Boudreaux E, Crawford S, et al. An exploration of the role of religion/spirituality in the promotion of physicians' wellbeing in Emergency Medicine. Preventive Medicine Reports. 2016;3:189-95 uses the 2-item MBI, which is controversial, rather than the full MBI, and while Roslan NS, Yusoff MSB, Ab Razak A, Morgan K, Shauki NIA, Kukreja A, et al. Training Characteristics, Personal Factors and Coping Strategies Associated with Burnout in Junior Doctors: A Multi-Center Study. HEALTHCARE. 2021;9(9) is a high quality paper, comparison between the Brief-COPE score and burnout score is not detailed within the paper, sadly. Within the limits of this manuscript, we have selected comparable studies that also look at doctors as their population, and use similar validated scores to those we used.

We agree that the statement as it stands is too strong, and the text now reads:

“The authors have found no published studies to date that measure burnout and spiritual health simultaneously in doctors using the validated measures used here (MBI and the FACIT-Sp-NI).”

---

## [Editor Report · Decision Letter 2]

13 Oct 2022

A National Study of Burnout and Spiritual Health in UK General Practitioners During the COVID-19 Pandemic

PONE-D-22-09960R2

Dear Dr. Whitehead,

We’re pleased to inform you that your manuscript has been judged scientifically suitable for publication and will be formally accepted for publication once it meets all outstanding technical requirements.

Kind regards,

Zhuo Chen, Ph.D.

Academic Editor

PLOS ONE

---

## [Editor Report · Acceptance letter]

24 Oct 2022

PONE-D-22-09960R2 

A National Study of Burnout and Spiritual Health in UK General Practitioners During the COVID-19 Pandemic 

Dear Dr. Whitehead:

I'm pleased to inform you that your manuscript has been deemed suitable for publication in PLOS ONE. Congratulations! Your manuscript is now with our production department. 

Kind regards, 

on behalf of

Prof. Zhuo Chen 

Academic Editor

PLOS ONE